# Quantifying Water Storage Changes and Groundwater Drought in the Huaihe River Basin of China Based on GRACE Data

Zunguang Zhou, Baohong Lu *, Zhengfang Jiang and Yirui Zhao

College of Hydrology and Water Resources, Hohai University, Nanjing 210098, China;
zhouzunguang@126.com (Z.Z.); 211301010015@hhu.edu.cn (Z.J.); ly847163563@gmail.com (Y.Z.)
* Correspondence: lubaohong@126.com; Tel.: +86-13851886088

**Abstract:** The Huaihe River Basin is an important ecological function conservation area in China, and it is also an important production area for national food, energy, minerals, and manufacturing. The groundwater storage and groundwater drought in this region are of great significance for ecological maintenance and water resources management. In this study, based on GRACE data and GLDAS data, a dynamic calculation method for groundwater storage in the Huaihe River Basin was developed, and a groundwater drought index (GRACE-GDI) was derived. By coupling GRACE-GDI with run theory, the quantitative identification of groundwater drought events, as well as their duration, intensity, and other characteristics within the basin, was achieved. The spatiotemporal changes in groundwater storage and groundwater drought in the Huaihe River Basin were analyzed using the developed method. The results showed that GRACE data are highly applicable in the Huaihe River Basin and is capable of capturing the spatiotemporal variations in groundwater storage in this region. Over the study period, mainly affected by rainfall, the terrestrial water storage and surface water storage in the Huaihe River Basin showed a decreasing trend, while groundwater storage showed a slight increasing trend. The duration of groundwater drought events in the basin ranged from 78 to 152 months, with an intensity of 82.77 to 104.4. The duration of drought gradually increased from north to south, while the intensity increased from south to north.

**Keywords:** GRACE data; drought index; trend analysis; water storage anomalies; groundwater drought

## 1. Introduction

Drought is a natural environmental disaster that intersects with various fields, including meteorology, environmental science, ecology, hydrology, agriculture, and geology, thereby attracting the attention of numerous scholars. Due to its extensive impact range, prolonged duration, and significant economic losses, drought adversely affects both economic growth and social stability [1]. Groundwater, a crucial component of the water cycle, plays a vital role in global and regional hydrology, climate change, and biogeochemical cycles. With its good water quality and stable supply conditions, groundwater serves as an essential source of freshwater for irrigated agriculture, industrial production, and urban life in many parts of the world [2].

Groundwater drought, a subset of hydrological drought, is primarily characterized by the sustained impact of reduced groundwater recharge or increased extraction, leading to a continuous decline in groundwater levels and a reduction in subsurface runoff [3]. This phenomenon can adversely affect residential water use, agricultural irrigation, and industrial production, as well as surface water bodies and ecosystems that depend on groundwater recharge. Unlike other forms of hydrological drought, groundwater drought is particularly susceptible to prolonged durations due to the extended time required for groundwater recharge and restoration [4]. Early stages of groundwater drought are often difficult to detect, thereby only garnering widespread attention when significant impacts

on human life are exhibited [5]. Persistent groundwater drought can deplete surface and groundwater resources, resulting in secondary disasters such as ground subsidence, soil salinization, and seawater intrusion [6].

The Gravity Recovery and Climate Experiment (GRACE) gravity satellite, developed in recent years, infers groundwater changes by observing variations in the Earth's gravity field, offering a new method for effectively monitoring global groundwater drought. This technology is particularly useful in regions with limited data and sparsely distributed monitoring stations, as it compensates for errors associated with interpolation using ground-based stations. Consequently, GRACE has been widely employed in groundwater storage and drought monitoring. Moghim et al. [7] used the GRACE satellite data to estimate water storage changes in Iran, finding that water storage in the northern regions exhibited significant fluctuations and a notable decreasing trend. Sediqi et al. [8] utilized the GRACE data to analyze changes in terrestrial water storage in Afghanistan and assessed the spatial distribution of water resource sustainability. Their study revealed that water resource sustainability was higher in the northeast and southwest regions, while it was lower in the south and central regions. Several scholars have leveraged the GRACE data to explore groundwater drought. For instance, Thomas et al. [9] assessed groundwater drought in California's Central Valley using the GRACE satellite data. They constructed a framework reflecting local groundwater drought conditions, considering both complex human activities and natural factors. Additionally, they proposed an improved total water storage anomaly index (MTSDI) based on GRACE observations, discovering that the drought events identified by MTSDI closely matched actual drought conditions. Hosseini-Moghari et al. [10] used the MTSDI to analyze drought conditions in the Markazi Basin of Iran from 2002 to 2016. The results showed that the MTSDI index demonstrated higher accuracy in detecting drought events compared to the commonly used drought monitoring indices, the standardized precipitation index (SPI) and the standardized precipitation evapotranspiration index (SPEI). The MTSDI records were highly consistent with other indices and exhibited a strong correlation, suggesting the significant potential of the GRACE data for developing drought monitoring capabilities. Kumar et al. [11] calculated the groundwater drought index (GGDI) for four basins in India from 2003 to 2016 and analyzed the characteristics of groundwater drought. The results indicated that the Cauvery River Basin experienced a severe drought from 2012 to 2015, with a duration of 42 months and a severity of $-27$. The Godavari River Basin showed a positive trend at both monthly and seasonal scales, while the other three basins exhibited an opposite trend. Many studies in China have also demonstrated GRACE's capability in drought monitoring, such as applications to the Yangtze River Basin [12], the North China Plain [13–15] and Southwest China [16,17]. However, research specifically focusing on groundwater drought remains relatively scarce.

However, there are still some problems to be solved in the application of GRACE data. The lack of some GRACE data and the low spatial resolution are the main problems faced by scholars in their research. The applicability of these data in small and medium-scale areas still needs further research. Therefore, it is necessary to adopt practical methods to improve the spatiotemporal resolution of GRACE data. The changes in groundwater reserves based on the water balance equation may have certain errors in humid and semi-humid areas and high mountainous plateau areas. In areas with severe water reserve losses, the drought index based on GRACE data can be more severe in the middle and late stages of the study period than at the beginning, but this is not caused by meteorological drought, but mostly due to human activities, such as excessive groundwater extraction and afforestation.

To address the general research gap on quantifying spatiotemporal variations in groundwater drought in the Huaihe River Basin, the first objective of this study was to couple GRACE data and the Global Land Data Assimilation System (GLDAS) to describe spatiotemporal variabilities in the groundwater storage of the Huaihe River Basin. The second objective was to explore the dynamics of groundwater drought in the Huaihe River Basin and their associated controls in relation to hydrometeorological conditions and basin characteristics.

## 2. Materials and Methods

### 2.1. Study Area

As shown in Figure 1, the Huaihe River Basin is located in eastern China, spanning from 111°55′ E to 121°20′ E and from 30°55′ N to 36°20′ N. The Huaihe River flows through the provinces of Henan, Anhui, and Jiangsu. Additionally, the basin includes parts of southern Shandong and northeastern Hubei provinces, covering a total area of 270,000 km². The basin is divided into two main river systems by the former course of the Yellow River: The Huaihe River system, covering 190,000 km², and the Yi-Shu-Si River system, covering 80,000 km². The Huaihe River Basin is situated in the transitional zone between northern and southern climates in China. The region north of the Huaihe River falls within the warm temperate zone, while the region south of the Huaihe River belongs to the northern subtropical zone. The climate is generally mild, with an annual average temperature ranging from 11 °C to 16 °C, increasing from north to south and from the coast to the inland areas. The long-term average annual precipitation in the Huaihe River Basin is approximately 920 mm, with rainfall decreasing from south to north. Precipitation is higher in mountainous areas than in plains, and greater along the coast than inland. The climate characteristics of the basin include dry and less rainy winters and springs, hot and rainy summers and autumns, and rapid transitions between cold and warm periods as well as between drought and flooding [18,19]

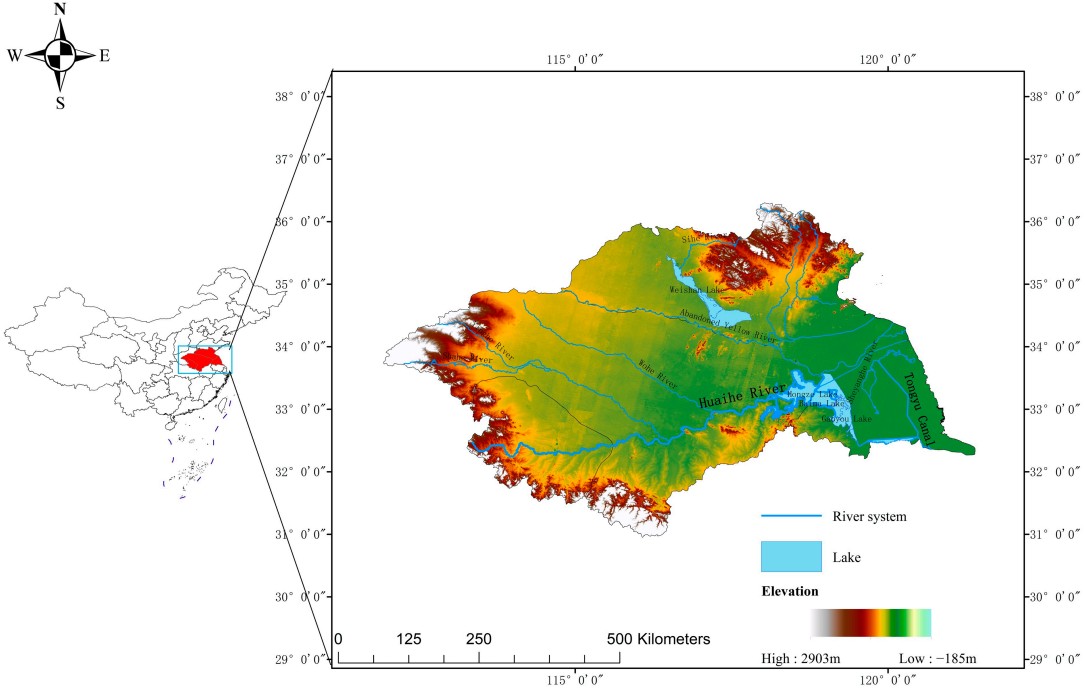

**Figure 1.** Location map of the study area (Huaihe River Basin).

### 2.2. Materials

#### 2.2.1. GRACE and GRACE FO RL06 Mascon Gravity Satellite Data

In this study, GRACE and GRACE-FO RL06 Mascon data products were utilized. The GRACE data are the Mascons provided by the Center for Space Research (CSR) at the University of Texas at Austin (http://www2.csr.utexas.edu/grace/, accessed on 1 April 2023). The GRACE Mascon product incorporates satellite laser ranging instead of C20 coefficients, first-order coefficient corrections, glacier equalization adjustments, and a specific correction factor for each grid. Compared to traditional spherical harmonic coefficients, the Mascon product, as a lattice data product, enhances signal resolution and reduces signal leakage error. Consequently, this study assessed water storage anomalies using GRACE and GRACE-FO RL06 Mascon data products to analyze drought change characteristics in the Huaihe

River Basin. The GRACE RL06 Mascon data covers the period from April 2002 to June 2017, while the GRACE-FO RL06 Mascon data spans from June 2018 to December 2020. Some of the missing time series were interpolated using cubic spline interpolation. Additionally, the data gap between July 2017 and May 2018, resulting from the transition between GRACE and GRACE-FO satellites, was addressed using a terrestrial water storage change dataset based on Chinese Regional Precipitation Reconstruction. This dataset, published by the National Tibetan Plateau Data Center, was reconstructed by establishing a precipitation reconstruction model that incorporates seasonal and trend terms of the Mascon data. This approach significantly improved data quality [20,21].

2.2.2. GLDAS-Noah Data

The Global Land Data Assimilation System (GLDAS) data (https://ldas.gsfc.nasa.gov/gldas/, accessed on 1 May 2023) are collaboratively developed by the Goddard Space Flight Center (GSFC) and the National Centers for Environmental Prediction (NCEP) using data assimilation techniques. This system integrates ground observations and satellite remote sensing data to drive land surface models. The data used in this study originates from GLDAS with a spatial resolution of $0.25° \times 0.25°$, specifically utilizing the output from the Noah model at a temporal resolution of one month. Data spanning from April 2002 to December 2022 were selected for analysis. Soil water content includes four layers extending to a depth of 2 m below the surface. Given that the Mascon data represent changes relative to the average value from January 2004 to December 2009, the GLDAS data were processed similarly. This involved subtracting the month-by-month surface water storage from the average surface water storage between January 2004 and December 2009 to calculate monthly changes in surface water storage.

*2.3. Methods*

2.3.1. Groundwater Storage Anomalies

Terrestrial water storage encompasses various components, including surface water, groundwater, soil water, snow and ice, biological water, and canopy water. However, some of these components are challenging to measure [22] and thus are not considered in the calculation [1,22]. Consequently, the groundwater storage anomalies can be determined using the following equation:

$$GWSA = TWSA - SMSA - SWESA - CWSA \qquad (1)$$

where *GWSA* is the groundwater storage anomalies; *TWSA* is the terrestrial water storage anomalies, obtained by inversion of GRACE satellite data, the mascon solutions with all the appropriate corrections applied (GAD, GIA, C20, C30, degree1, etc.) in equi-angular grid, the product requires no prior data preprocessing; *SMSA* is the soil water storage anomalies; *SWESA* is the snow water equivalent anomalies; *CWSA* is the plant canopy water storage anomalies. The units of all variables in Equation (1) are centimeters. *SMSA*, *SWESA*, and *CWSA* were obtained from the GLDAS Noah model.

2.3.2. Calculation of Groundwater Drought Index Based on GRACE Data

Zhao et al. [23] proposed a drought index called the GRACE drought severity index (GRACE-DSI), which is based on the inversion of terrestrial water storage anomalies derived from the GRACE data. This index provides a novel approach for evaluating regional hydrological drought in areas with limited measured data. Additionally, Chu [1] applied this method in Northwest China to calculate the groundwater drought index (GRACE-GDI) and conducted drought studies. This study used the method of Chu [1] to compute the drought index:

$$GRACE - GDI_{i,j} = \left( GWSA_{i,j} - \overline{GWSA_j} \right) / \sigma_j \qquad (2)$$

where *i* and *j* are the year and month, respectively; $GWSA_{i,j}$ is the groundwater storage anomalies in month *j* of year *i*; $\overline{GWSA_j}$ and $\sigma_j$ are the mean and standard deviation of the groundwater storage anomalies in month *j*, respectively.

When *GRACE-GDI* $\leq -2.0$, it indicates that extreme groundwater drought occur in that month. When $-2.0 < $ *GRACE-GDI* $\leq -1.5$, it is classified as severe groundwater drought. When $-1.5 < $ *GRACE-GDI* $\leq -1.0$, it indicates moderate groundwater drought. When $-1.0 < $ *GRACE-GDI* $\leq -0.5$, it is considered mild groundwater drought. If *GRACE-GDI* $> -0.5$, it indicates that no drought event occurs during that month.

### 2.3.3. Theil-Sen Slope Estimation

The Theil-Sen slope method, a nonparametric test introduced and developed by Sen [24], is widely used to estimate trend changes in time series. Nan et al. [25] utilized this method to analyze the spatiotemporal distribution characteristics of precipitation in the Chongqing area from 1965 to 2014. Yuan et al. [26] analyzed the spatiotemporal variations in vegetation cover in the Yellow River Basin from 2000 to 2010. The slope was calculated using the following formula:

$$Q_i = \left( x_j - x_k \right) / (j - k) \quad i = 1, 2, \cdots, N \tag{3}$$

where $x_j$, $x_k$ are the time series values of the $j$th and $k$th samples, respectively, with $j > k$ and $N = n(n - 1)/2$.

Arranging the $Q_i$ values from smallest to largest, the median Theil-Sen slope is:

$$Q_{med} = \begin{cases} Q_{(N+1)/2} & N \text{ is an odd number} \\ \left( Q_{(N+2)/2} + Q_{N/2} \right) / 2 & N \text{ is an even number} \end{cases} \tag{4}$$

$N$ represents the total number of data points for each grid cell. The metric $Q_{med}$ reflects the degree of skewness in the time series trend. A value greater than 0 indicates an upward trend in the sample, while a value less than 0 signifies a downward trend.

### 2.3.4. Mann-Kendall Trend Test

The Mann-Kendall trend test [27] was employed to assess the significance of the temporal trend in groundwater storage in the Huaihe River Basin. Pathak et al. [28] analyzed groundwater levels in the Ghataprabha Basin of India and assessed regional groundwater drought. Thomas et al. [29] used GRACE data and applied the Mann-Kendall (M-K) trend analysis to examine groundwater depletion caused by climate factors.

The test results were categorized into four grades based on significance: (1) extremely significant change with $|Z| > 2.58$; (2) significant change with $1.96 < |Z| \leq 2.58$; (3) weakly significant change with $1.65 < |Z| \leq 1.96$; (4) no significant change with $0 < |Z| \leq 1.65$.

### 2.3.5. Drought Identification

The run theory is a method for analyzing time series. It has been applied to which can be applied to drought event recognition, such as applications in Loess Plateau [30] and China [31]. In this study, multi-threshold run analysis was conducted to identify drought. The core of the multi-threshold method is to use multiple different thresholds to segment and intercept the time series, and then extract and analyze the drought characteristic variables. In practice, there are many ways to select the number of thresholds and different thresholds. In this study, the three-threshold drought identification method [32] was used. As shown in Figure 2, $X$ represents the corresponding drought index. The first step requires setting three different drought indices in advance: $X_1$, $X_0$, and $X_2$, with $X_1 > X_0 > X_2$. In the second step, drought identification is carried out for the time series $X$ using $X_0$, and four initial droughts (a, b, c, and d) can be identified. In the third step, if the drought events b and c are adjacent to each other with a one-month interval, they should be merged into one drought event. If the value of $X$ between the two drought events is less than $X_1$, b and c are merged into one drought event. Otherwise, they remain separate. In the fourth step, drought events with a duration of only one month are analyzed. If $X$ is smaller than $X_2$, it is considered a drought event with a one-month duration. Otherwise, it is not considered a drought event. Consequently, a is retained, and d is deleted. Through these steps, two

drought events, a and b + c, are identified from the time series *X*, and their corresponding drought durations and intensities can be calculated and extracted. The three-thresholds selected in this study are: $X_1 = 0.5$, $X_0 = 0$, and $X_2 = -0.5$.

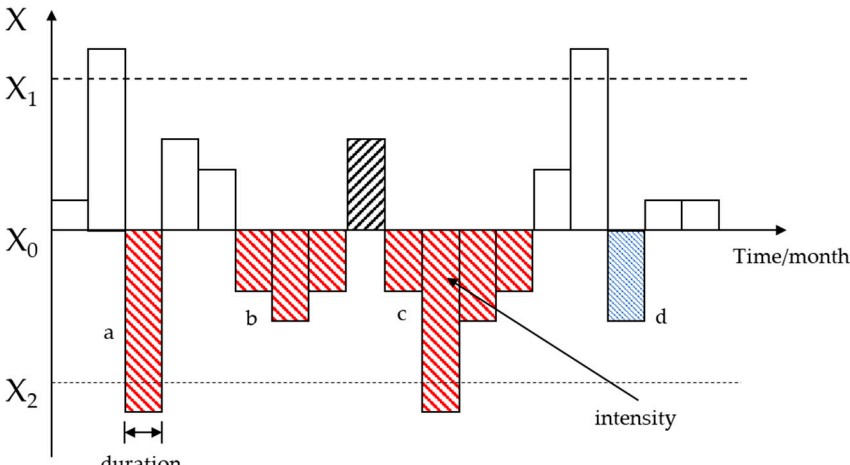

**Figure 2.** Identification of drought variables through the run theory with multiple thresholds. (The horizontal axis is time, the vertical axis is the drought index GRACE-GDI, duration represents the duration of a drought from the beginning to the end, intensity is the area of red shadow, $X_1$, $X_2$ and $X_0$ represent the threshold values of the three GRACE-GDI).

## 3. Results and Analysis

### 3.1. Trend Analysis of Terrestrial and Surface Water Storage Anomalies

3.1.1. Trend of Terrestrial Water Storage Anomalies

Figure 3 illustrates the trend of terrestrial water storage in the Huaihe River Basin based on the M-K trend test. The results indicated that terrestrial water storage anomalies generally exhibited a decreasing trend across most of the basin, affecting 86.87% of the area. Within this region in a decline trend, significant and weakly significant decreases accounted for 4.12% and a smaller portion of the basin area, respectively. In contrast, 12.17% of the basin area showed no significant change, with these regions primarily situated in the southern and eastern parts of the basin. Conversely, some areas of the Huaihe River Basin, particularly in the southern and eastern regions, displayed an increasing trend in water storage. This increasing trend affected only 0.96% of the basin area. Within this small region, significant and weakly significant increases each represented 50% of the area with an upward trend.

As shown in Table 1, the areas experiencing a decrease in terrestrial water storage were predominantly located in the upper, middle, and Yi-Shu-Si River basins, comprising 77.55%, 90.86%, and 100% of these regions, respectively. Conversely, areas with no significant change were distributed across the upper, middle, and lower reaches, representing 22.45%, 9.14%, and 44% of their respective areas, with the lower reaches having a higher proportion of such areas. Increasing trends in water storage were observed primarily in the downstream regions, accounting for only 8% of the downstream area.

Figure 4 illustrates the spatial distribution of the magnitude of change in terrestrial water storage anomalies across the Huaihe River Basin. The figure reveals that 4.8% of the basin area had a magnitude of change greater than 0, with all rates ranging between 0 and 0.26 cm/year. Conversely, 2.16% of the area exhibited a magnitude of change less than −2 cm/year, 15.59% fell between −2 and −1 cm/year, and 77.46% was between −1 and 0 cm/year. This indicates that 95.2% of the basin area has a magnitude of change less than 0 cm/year. Specifically, 87.24% of the regions showed a magnitude of change between −1 and 0 cm/year, reflecting a slow decrease in surface water storage in the majority of these areas. In contrast, regions with an increasing trend all exhibited a magnitude of change between 0 and 0.26 cm/year, indicating a very gradual increase in surface water storage.

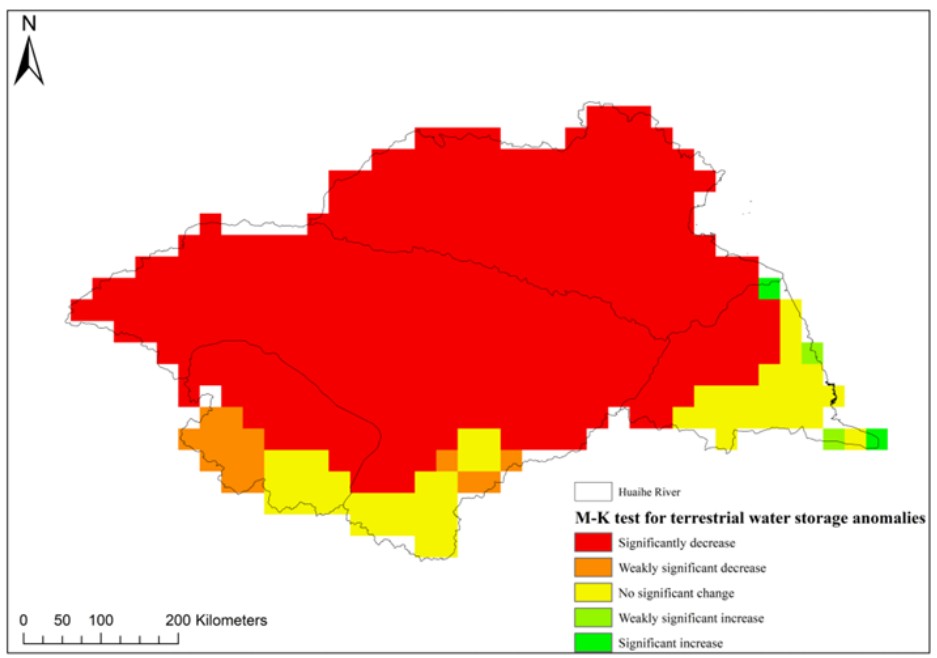

**Figure 3.** Trends in terrestrial water storage anomalies.

**Table 1.** Statistics on the number of zonal grids for trends in terrestrial water storage.

| Trend | Upstream | Middlestream | Downstream | Yi-Shu-Si River | Total |
|---|---|---|---|---|---|
| Significant decrease | 27 | 175 | 24 | 123 | 349 |
| Weakly significant decrease | 11 | 4 | 0 | 0 | 15 |
| No significant change | 11 | 18 | 22 | 0 | 51 |
| Weakly significant increase | 0 | 0 | 2 | 0 | 2 |
| Significantly increased | 0 | 0 | 2 | 0 | 2 |
| Total | 49 | 197 | 50 | 123 | 419 |

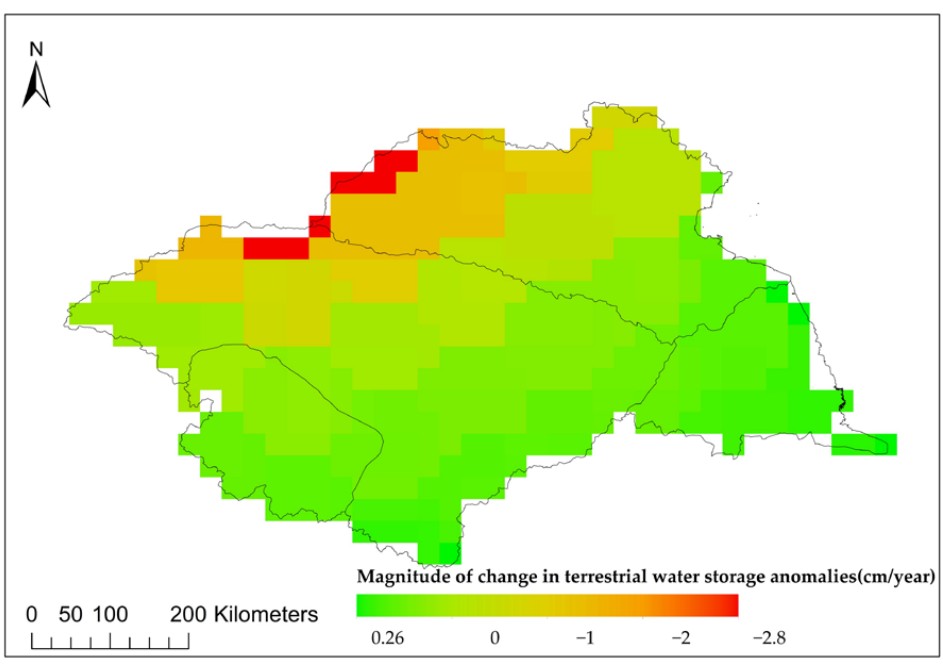

**Figure 4.** Magnitude of change in terrestrial water storage anomalies.

### 3.1.2. Trend of Surface Water Storage Anomalies

Figure 5 presents the results of the Mann-Kendall trend test for surface water storage anomalies in the Huaihe River Basin. The analysis indicated that 53.7% of the basin area showed no significant change in surface water storage anomalies. Areas exhibiting a decreasing trend account for 29.12% of the basin area, with 75.41% of this region showing significant decreases and 24.59% showing weakly significant decreases. Conversely, 17.18% of the basin area displayed an increasing trend in surface water storage. Within this area, 40.28% showed significant increases, while 59.72% showed weakly significant increases. This indicated that the regions with increasing water storage trends constituted 17.18% of the basin area, with a notable proportion demonstrating varying levels of statistical significance.

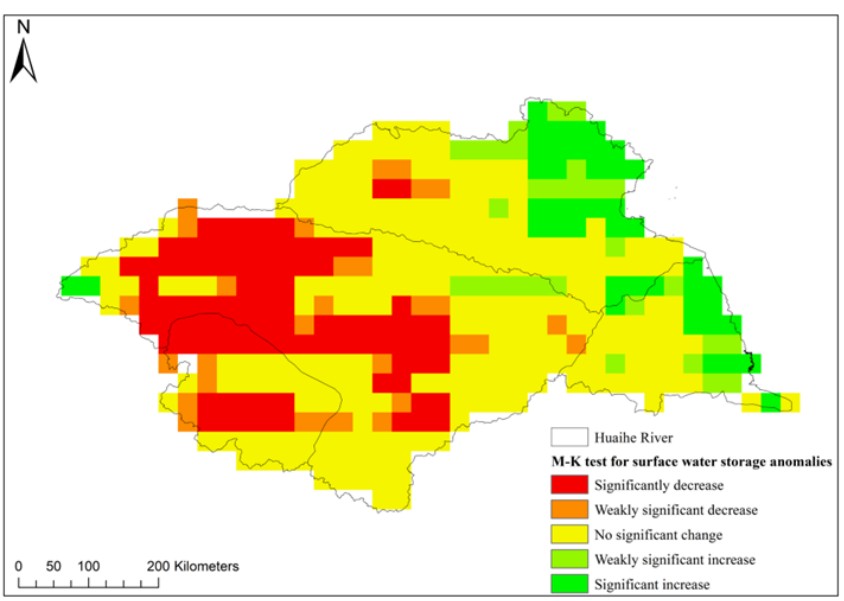

**Figure 5.** Trends in surface water storage anomalies.

As detailed in Table 2, the areas experiencing a decrease in surface water storage were primarily concentrated in the upper and middle reaches, comprising 53.07% and 45.68% of these respective areas. Regions with no significant change were distributed throughout the entire basin, representing 46.94% of the upper reaches, 50.76% of the middle reaches, 56% of the lower reaches, and 60.16% of the Yi-Shu-Si River Basin. The areas showing an increase in surface water storage were predominantly located in the lower reaches and the Yi-Shu-Si River Basin, accounting for 44% and 34.96% of these respective regions.

**Table 2.** Statistics on the number of zonal grids for surface water storage trends.

| Trend | Upstream | Middlestream | Downstream | Yi-Shu-Si River | Total |
|---|---|---|---|---|---|
| Significant decrease | 19 | 71 | 0 | 2 | 92 |
| Weakly significant decrease | 7 | 19 | 0 | 4 | 30 |
| No significant change | 23 | 100 | 28 | 74 | 225 |
| Weakly significant increase | 0 | 5 | 8 | 16 | 29 |
| Significantly increased | 0 | 2 | 14 | 27 | 43 |
| Total | 49 | 197 | 50 | 123 | 419 |

Figure 6 illustrates the spatial distribution of the magnitude of change of surface water storage in the Huaihe River basin. The magnitude of change in the basin was as follows: greater than 3 cm/year, accounting for 0.96%; between 2 and 3 cm/year, accounting

for 4.57%; between 1 and 2 cm/year, accounting for 15.63%; and between 0 and 1 cm/year, accounting for 23.08%. The total area with a magnitude of change greater than 0 cm/year was 44.23%. On the other hand, the magnitude of change in the basin was less than −4 cm/year, accounting for 0.72%; between −4 and −3 cm/year, accounting for 0.72%; between −3 and −2 cm/year, accounting for 23.08%. The rate less than −4 cm/year accounted for 0.72%; that between −4 and −3 cm/year accounted for 5.29%; that between −3 and −2 cm/year accounted for 13.94%; that between −2 and −1 cm/year accounted for 14.42%; and that between −1 and 0 cm/year accounted for 21.39%. The total area of the magnitude of change less than 0 cm/year was 55.77%. Overall, most areas in the basin experienced a magnitude of change less than 0 cm/year, indicating a decreasing tendency for surface water storage. Additionally, 69.05% of the areas had a magnitude of change between 0 and −2 cm/year, suggesting that most of the reduced areas had a slower decrease rate in surface water storage. Conversely, in areas where the magnitude of change was greater than 0 cm/year, 87.52% of the areas had a magnitude of change between 0 and 2 cm/year, indicating a slower increase rate in the most regions with an increasing trend.

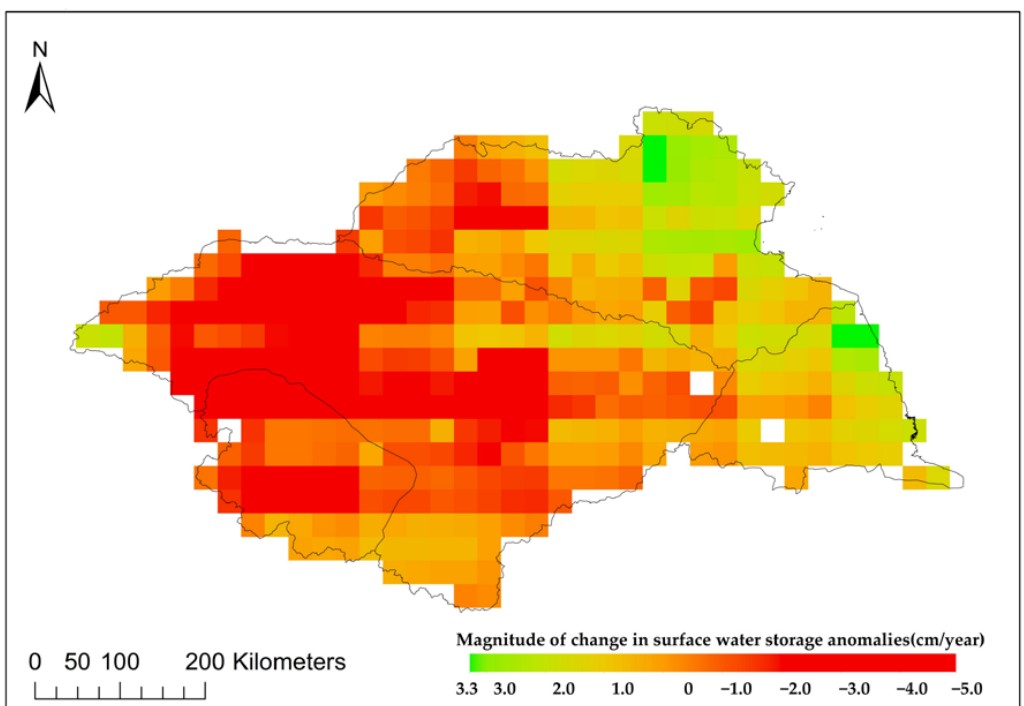

**Figure 6.** Magnitude of change in surface water storage anomalies.

*3.2. Calculation and Trend Analysis of Groundwater Storage Anomalies*

3.2.1. Calculation and Verification of Groundwater Storage Anomalies

Calculation of Groundwater Storage Anomalies

Figure 7 provides insights into the changes in water storage within the Huaihe River Basin. Figure 7a shows the average terrestrial water storage changes over time. The analysis revealed an overall decreasing trend in terrestrial water storage, with an annual magnitude of change of −7.5 cm/year according to least-squares fitting. The highest recorded change in terrestrial water storage anomalies was 10.26 cm in September 2005, while the lowest was −22.56 cm in March 2020. The average monthly change in terrestrial water storage anomalies peaked at 38.75 cm in July and reached its minimum at 17.65 cm in January. Seasonal variations showed a maximum average change of 39.66 cm in summer and a minimum of 22.38 cm in winter. Figure 7b illustrates groundwater storage changes. This figure demonstrates an upward trend in groundwater storage, with an annual magnitude of change of 4.02 cm/year based on least-squares fitting. The highest change in groundwater storage anomalies was 16.40 cm in October 2017, and the lowest was −16.58 cm

in February 2013. The average monthly change in groundwater storage anomalies was highest in August at 27.7 cm and lowest in April at 17.08 cm. The seasonal average change was greatest in autumn at 24.12 cm and smallest in winter at 18.55 cm. Overall, while terrestrial water storage showed a declining trend, groundwater storage demonstrated an increasing trend over the observed period.

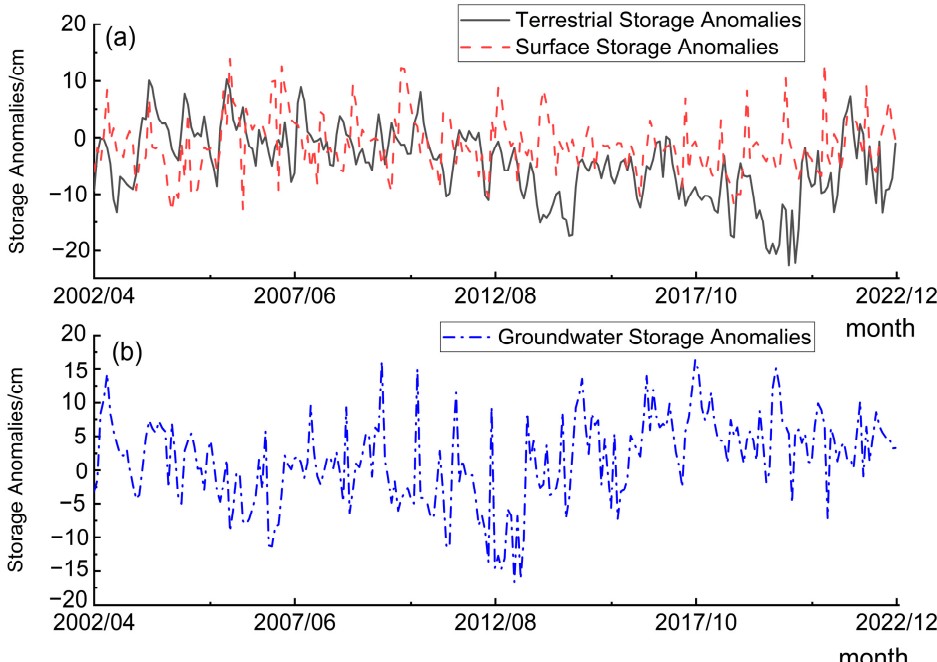

**Figure 7.** Mean value of water storage anomalies in the basin. (**a**) shows the average of land and surface storage anomalies, while (**b**) shows the average of groundwater reserves storage anomalies.

Verification of Groundwater Storage Anomalies

Due to the fact that changes in groundwater storage can be obtained by multiplying water level changes by specific yield, water level dynamics can be used to characterize the groundwater storage changes. As GRACE and GLDAS data reflect average conditions at a grid scale, while well observation data reflect water level changes at specific points, the results have a certain degree of uncertainty. However, at a large regional scale, this method and its associated results can reflect the overall trend and changing characteristics of groundwater storage. Therefore, validation of groundwater storage anomalies was carried out using collected monthly groundwater level data from the Huaihe River Basin. The process involved extracting groundwater storage change values from the GRACE Mascons data corresponding to the latitude and longitude of the monitoring stations. These extracted data were then compared to the measured groundwater level data to determine the correlation between the two datasets. Table 3 presents the Pearson correlation coefficients between the measured groundwater level data and the GRACE-derived groundwater storage changes. The results indicated a generally strong correlation, with coefficients reaching up to 0.69. However, some stations exhibited lower correlation coefficients, which may be attributed to factors such as excessive groundwater extraction due to human activities. Figure 8 illustrates the consistency between groundwater storage changes derived from the GRACE Mascons data and the trend observed in the measured groundwater levels. This visual representation confirms that the GRACE-derived groundwater storage anomalies closely aligned with the actual changes in groundwater storage as recorded by the monitoring stations. The consistency between the GRACE data and the measured data confirmed the high accuracy of the GRACE satellite data in reflecting groundwater storage variations in the Huaihe River Basin.

**Table 3.** Correlation coefficients between measured groundwater level data and groundwater storage anomalies.

| Station Code | Station Name | Longitude | Latitude | Correlation Coefficient |
|---|---|---|---|---|
| 40177397 | Guo Yukai Lankao No. 2 | 114.7651° E | 34.82942° N | 0.69 |
| 50261127 | Guo Yuzhu Queshan No. 2 | 114.0482° E | 32.59691° N | 0.57 |
| 50261551 | Guo Yuzhu Zhengyang No. 2 | 114.3159° E | 32.47336° N | 0.66 |
| 50261869 | Guo Yuzhu Zhengyang No. 7 | 114.3269° E | 32.38529° N | 0.61 |
| 50370040 | Aiting | 115.2387° E | 32.68268° N | 0.61 |
| 50465040 | Huoqiu | 116.2962° E | 32.36457° N | 0.45 |
| 50465080 | Hongji | 116.1771° E | 31.90755° N | 0.45 |
| 50971160 | Qiuji | 118.314° E | 32.83671° N | 0.36 |
| 50971200 | Taiping | 118.4673° E | 33.52594° N | 0.48 |
| 51162049 | Xiaogu Town | 119.4256° E | 35.33786° N | 0.45 |
| 51162073 | Taoge | 119.3772° E | 35.28697° N | 0.68 |

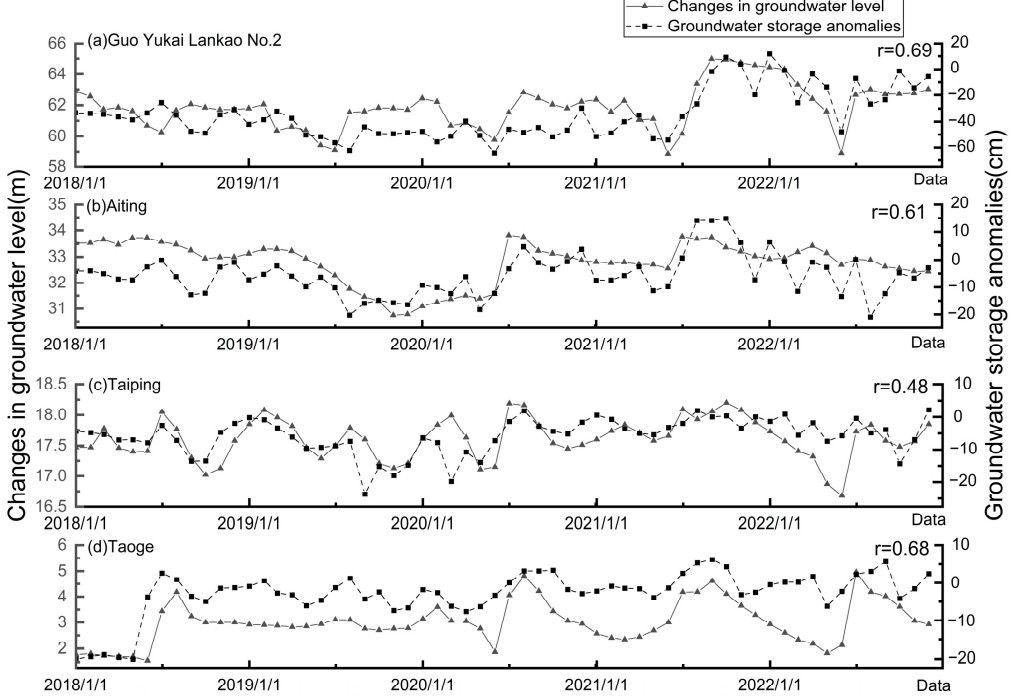

**Figure 8.** Correlation between groundwater storage changes and measured groundwater levels.

### 3.2.2. Trend Analysis of Groundwater Storage Anomalies

Figure 9 displays the results of the Mann–Kendall trend test for groundwater storage anomalies in the Huaihe River Basin. The figure indicates a predominance of increasing trends in groundwater storage across the basin. Specifically, 77.33% of the basin area demonstrated an increasing trend. Of the increasing area, 89.2% showed a statistically significant increase, while 10.8% exhibited a weakly significant increase. Conversely, 22.67% of the basin area showed no significant change in groundwater storage anomalies. This overall trend suggested that groundwater storage generally increased in the Huaihe River Basin, with significant increases observed in the majority of the region.

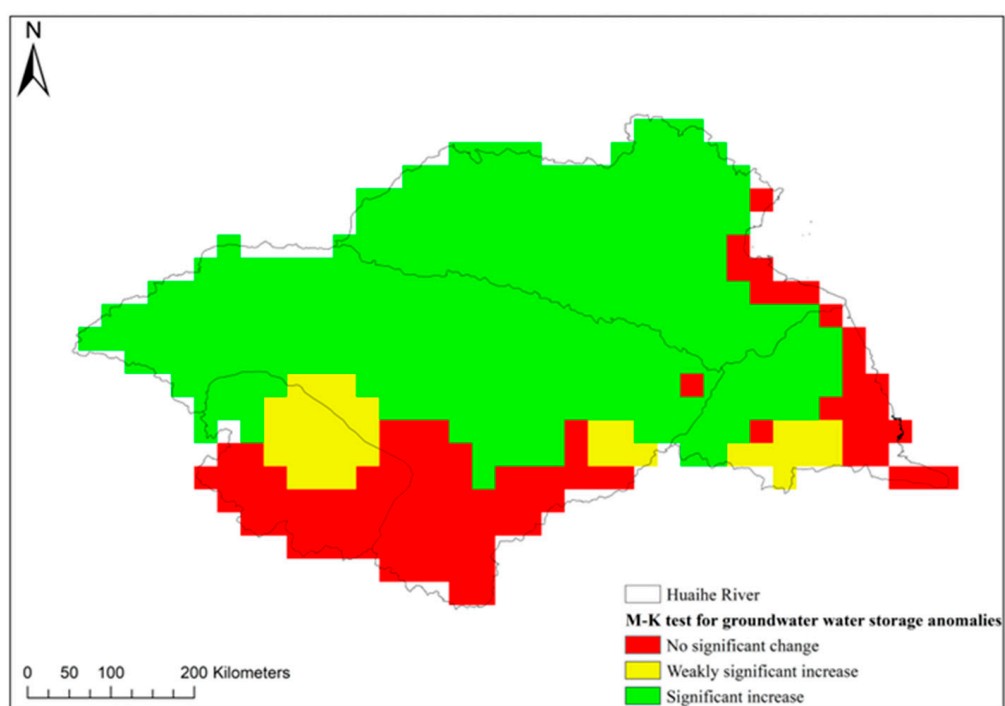

**Figure 9.** Trends in groundwater storage anomalies.

Table 4 details the distribution of groundwater storage anomalies within the Huaihe River Basin. It shows that areas with no significant change were widespread, representing 53.06%, 22.84%, 34%, and 5.69% of the upper, middle, lower, and Yi-Shu-Si River basins, respectively. Conversely, the areas exhibiting an increase in groundwater storage were predominantly located in the middle and Yi-Shu-Si River basins, accounting for 77.16% and 94.31% of the area in the lower and Yi-Shu-Si River basins, respectively. This distribution highlighted a significant regional variation in groundwater storage trends, with notable increases primarily concentrated in the middle and Yi-Shu-Si River basins.

**Table 4.** Statistics on the number of grids in the zones of groundwater storage changes.

| Trend | Upstream | Middlestream | Downstream | Yi-Shu-Si River | Total |
|---|---|---|---|---|---|
| No significant change | 26 | 45 | 17 | 7 | 95 |
| Weakly significant increase | 16 | 10 | 9 | 0 | 35 |
| Significantly increased | 7 | 142 | 24 | 116 | 289 |
| Total | 49 | 197 | 50 | 123 | 419 |
| Significantly increased | 0 | 0 | 2 | 0 | 2 |
| Total | 49 | 197 | 50 | 123 | 419 |

Figure 10 depicts the spatial distribution of groundwater storage change rates in the Huaihe River Basin. The figure illustrates that the magnitude of change varied between −0.05 and 0.94 cm/year, indicating a very gradual trend in both increasing and decreasing groundwater storage across the basin. Notably, 98.3% of the basin area exhibited a magnitude of change greater than 0 cm/year, with values ranging from 0 to 0.94 cm/year. In contrast, only 1.7% of the basin area showed a magnitude of change less than 0 cm/year, with values ranging between −0.05 and 0 cm/year. This distribution suggested that groundwater storage across the basin generally increased, although the increase rate was very modest.

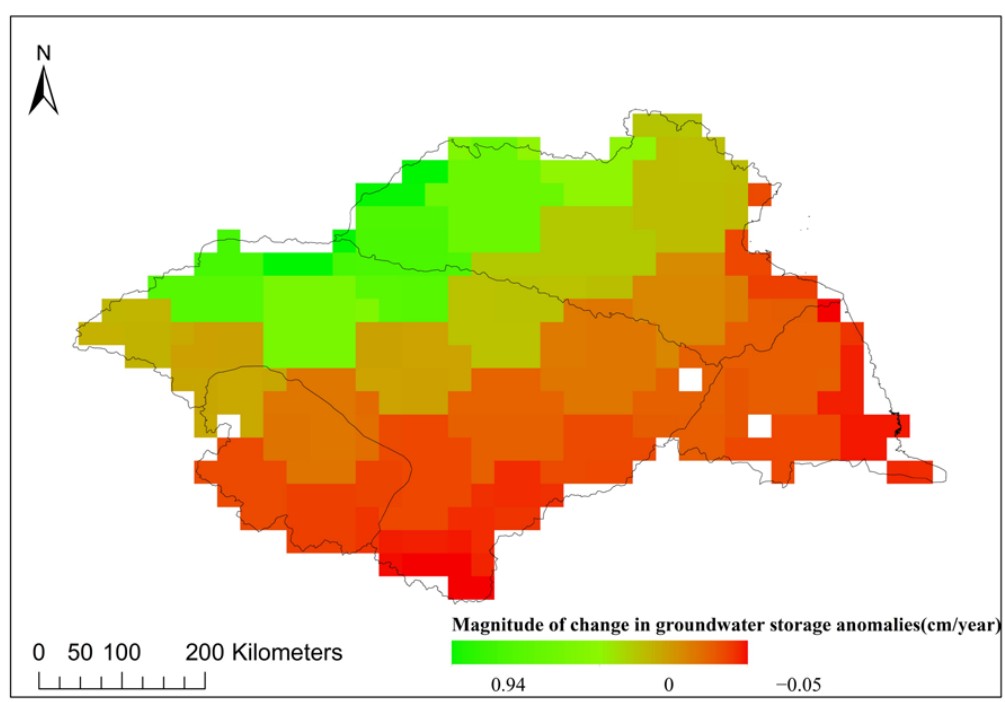

**Figure 10.** Magnitude of change in groundwater storage anomalies.

*3.3. Groundwater Drought Characteristics*

3.3.1. Drought Frequency, Total Duration, and Total Intensity

Figure 11 illustrates the spatial distributions of groundwater drought frequency, total duration, and total intensity in the Huaihe River Basin from April 2002 to December 2022.

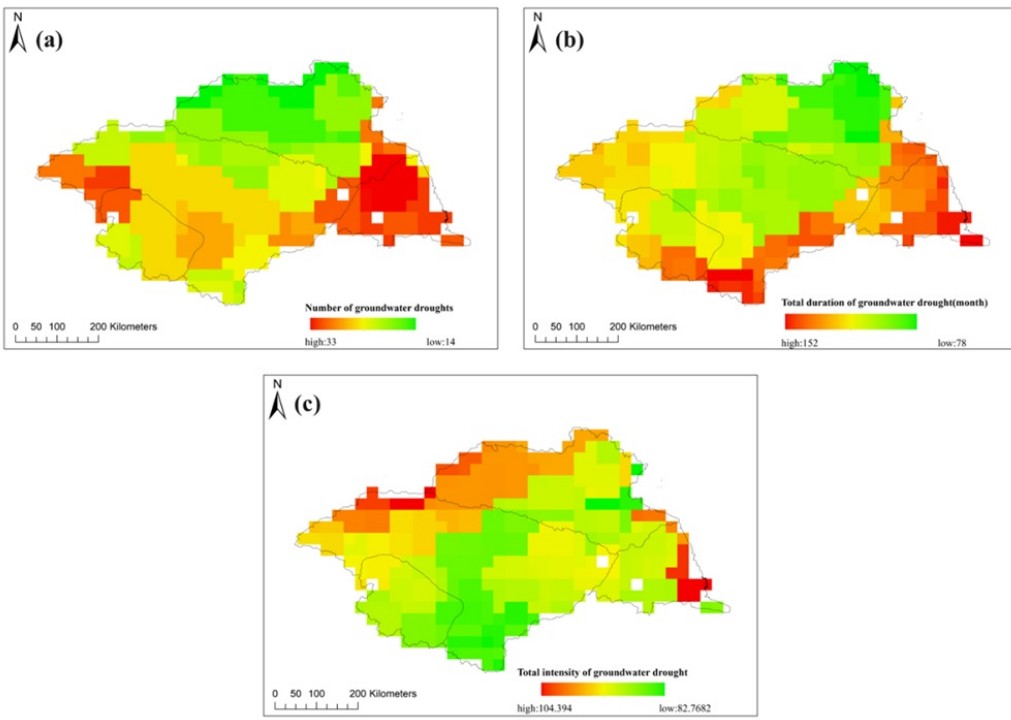

**Figure 11.** Groundwater drought characterization variables. (**a**) shows the total number of groundwater droughts, (**b**) shows the total duration of groundwater droughts, and (**c**) shows the total intensity of groundwater droughts.

As shown in Figure 11a, the frequency of groundwater drought events in the basin ranged from 14 to 33 occurrences. The distribution revealed a gradual increase in drought frequency from north to south. The middle and lower reaches of the basin experienced a higher frequency of droughts, whereas the Yi-Shu-Si River Basin exhibited relatively fewer drought events. Areas with more than 23 drought occurrences during the study period were common, with some regions experiencing up to 30 droughts. This high frequency of groundwater droughts exacerbated local water resource issues and posed risks to ecological and environmental health, including soil degradation and reduced vegetation cover.

Figure 11b illustrates the total duration of groundwater droughts, which varied from 78 to 152 months, accounting for approximately 31.32% to 61.04% of the total study period of 249 months. The duration of droughts at the grid level within the basin showed significant variation. Nearly half of the regions experienced droughts lasting over 110 months, with substantial portions of these areas being under drought conditions for almost half of the study period. The total duration of groundwater droughts increased from north to south, following a similar spatial pattern to drought frequency. Longer drought durations were primarily observed in the lower reaches, southern, and western parts of the middle reaches, whereas the Yi-Shu-Si River Basin experienced shorter drought durations.

Figure 11c depicts the groundwater drought intensity, which ranged from 82.77 to 104.40. The intensity showed a gradual increase from the southern to the northern regions of the basin. About half of the basin experienced drought intensities between 93 and 104. Notably high drought intensities were observed in the western section of the middle reaches, the eastern part of the lower reaches, and the western portion of the Yi-Shu-Si River Basin.

### 3.3.2. Maximum Duration and Intensity

Figure 12 illustrates the spatial distribution of the maximum duration and intensity of single groundwater drought events in the Huaihe River Basin.

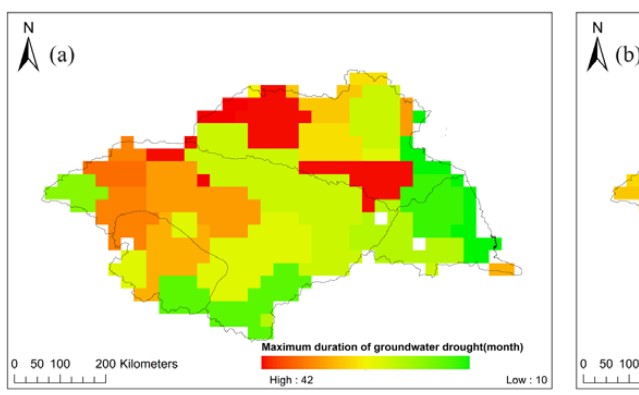 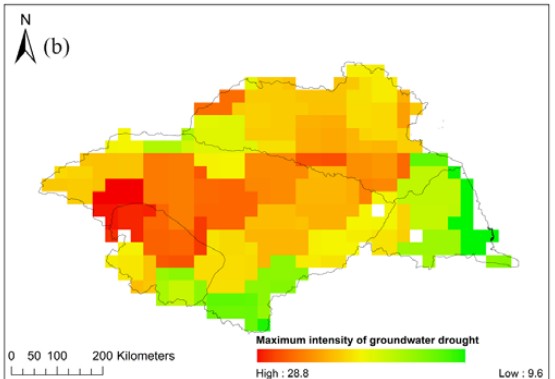

**Figure 12.** Maximum duration and intensity of groundwater drought. (**a**) shows the maximum duration of groundwater drought, and (**b**) shows the maximum intensity of groundwater drought.

Figure 12a shows that the maximum duration of a single groundwater drought event ranged from 10 to 42 months. The spatial variation in maximum drought duration was significant. Areas with drought durations exceeding 40 months accounted for 1.21% of the basin, those with durations between 30 and 40 months represented 0.24%, durations from 20 to 30 months covered 38.41%, and durations from 10 to 20 months comprised 60.14%. Most areas experienced a maximum drought duration between 10 and 20 months, with the western and northern parts of the basin exhibiting durations exceeding 20 months.

Figure 12b presents the maximum intensity of a single drought event, which ranged from 9.63 to 28.80. The spatial distribution of maximum drought intensity was also notable. Areas with an intensity greater than 20 account for 65.94% of the basin, while those with intensity less than 20 made up 34.06%. Lower maximum intensity areas were predominantly located in the eastern and southern parts of the basin.

### 3.3.3. Average Drought Duration and Average Intensity

Figure 13 illustrates the average duration and intensity of groundwater droughts across the Huaihe River Basin.

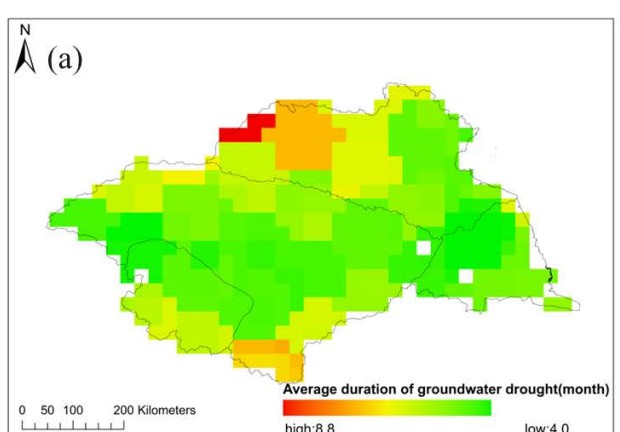 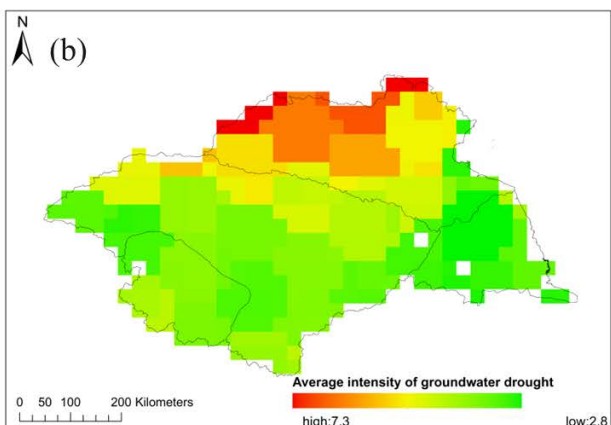

**Figure 13.** Average duration and intensity of groundwater droughts. (**a**) shows the average duration of groundwater drought, and (**b**) shows the average intensity of groundwater drought.

In Figure 13a, the average drought duration ranged from 4.03 to 8.79 months. The data showed a decreasing trend from north to south, with shorter durations in the southern and central regions of the basin and in the western Yi-Shu-Si River Basin.

Figure 13b indicates that the average drought intensity ranged from 2.81 to 7.27, also decreasing from north to south. The Yi-Shu-Si and Surabaya River basins display relatively higher average drought intensity. The spatial distribution of the duration and intensity of groundwater drought were similar, indicating a close relationship between the two. Prolonged drought often leads to an increase in its intensity. This poses significant challenges for regional ecological and water resources management.

## 4. Discussion

Changes in groundwater storage anomalies directly affect the occurrence and characteristics of groundwater drought, forming a negative feedback loop. The reduction in groundwater storage exacerbates the frequency and duration of drought, and the occurrence of groundwater drought will further weaken the recovery capacity of groundwater storage.

### 4.1. Water Storage Anomalies

During the study period, both land water storage and surface water storage exhibited a decreasing trend. The most probable explanation for this trend is the relatively low rainfall in the study area during this period. According to the China Water and Drought Bulletin, the Huaihe River Basin experienced five relatively severe drought events during the study period. For example, the summer drought in 2014 was particularly severe, accompanied by prolonged high temperatures, leading to a decrease in water levels in the upper and middle reaches of the Huaihe River. Influenced by the high temperatures and relatively low rainfall, the basin experienced strong evapotranspiration, exacerbating the drought. This finding aligned with the research of Zhao et al. [33] and Li et al. [34]. Precipitation, runoff, evapotranspiration, and human activities were found to influence terrestrial water storage. Wu et al. [35] demonstrated that precipitation and runoff were the primary factors affecting changes in terrestrial water storage, with a positive correlation between these variables and water storage trends. Moreover, increasing evapotranspiration led to a more rapid decline in water storage. Additionally, Feng et al. [36] highlighted that human activities significantly impacted water storage. The high population density in the Huaihe River Basin has led to overexploitation of water resources to meet various demands, further contributing to the observed decline in terrestrial water storage.

The overall trend of groundwater storage in the Huaihe River Basin exhibited a modest increasing trend, distinct from the changing trends of terrestrial water and surface water storage. This finding was consistent with that of Zhou et al. [37], though their study focused solely on the period from 2003 to 2009. In contrast, Wang et al. [38] extended their analysis of groundwater storage trends in the Huang-Huai-Hai region from 2003 to 2021, revealing an increase in groundwater storage in the upper and middle reaches of the Huaihe River Basin, while changes in the lower reaches remained relatively insignificant. Precipitation infiltration was identified as the most direct factor influencing changes in groundwater reserves. The Huaihe River Basin, characterized by abundant precipitation and significant inter-annual variability, provides conditions conducive to groundwater recharge, facilitating the recovery of groundwater reserves. As a result, several studies have demonstrated a notable correlation between groundwater reserve changes and precipitation trends.

### 4.2. Drought Characteristics

In general, groundwater drought in the Huaihe River Basin was characterized by prolonged durations, low intensity, and minimal disaster impact in most regions. However, some areas in the northern part of the basin experienced groundwater droughts with shorter duration but higher intensity. Groundwater drought events developed relatively slowly, making them challenging to detect and address. Consequently, groundwater droughts were prone to evolving into long-term and high-intensity disasters. Meteorological drought, being the initial phase of the drought transmission chain, often preceded other types of droughts, which typically lagged behind meteorological drought. Yan et al. [39] analyzed meteorological drought characteristics in the Huaihe River Basin using SPI. Their findings indicated that the spatial distribution of drought frequency mirrored that of groundwater drought and was generally greater. However, there was a discrepancy in the spatial distribution of the average duration and intensity of the two types of droughts, with groundwater drought exhibiting greater average duration and intensity compared to meteorological drought. Therefore, while meteorological drought is strongly associated with groundwater drought, the duration and intensity of groundwater drought generally exceed those of meteorological drought.

### 4.3. Limitations and Recommendations

This study has several limitations. (1) Human activities, e.g., coal mining, reservoir storage, large-scale construction, and inter-basin water transfers, may impact the Earth's gravitational field, thereby influencing the accuracy of the results. Additionally, the limited number of observation wells (11 in total) used to validate the calculated groundwater storage in this study may also affect the accuracy of the results. (2) Whist the results here need further assessment and testing requires longer and more detailed (e.g., better characterization of human actions) observation data, it is an encouraging step forward in groundwater drought estimation in large river basins. (3) Based on the spatial resolution of GRACE and GLDAS data, the grid resolution used for computation and analysis in this study was $0.25° × 0.25°$. Consequently, the spatial distribution maps of related groundwater storage and drought analysis exhibit a jagged appearance. However, at large-scale basins such as the Huaihe River Basin, this resolution can capture the spatial distribution of relevant information in the study area, and has a relatively minimal impact on the overall accuracy of the results [40].

### 5. Conclusions

In this study the characteristics of groundwater drought in the Huaihe River Basin was investigated. Due to the difficulty in obtaining large-scale groundwater data, the GRACE data were used to retrieve changes in terrestrial water storage. Changes in surface water storage were obtained from the GLDAS data, which were then used to derive groundwater storage variations. Qualitative analysis was conducted on the trends of terrestrial, surface,

and groundwater storage changes. A drought index was applied to study the characteristics of groundwater drought. The main findings are as follows:

The majority of the basin showed a decreasing trend in terrestrial water storage, covering 86.87% of the basin area, while about 77.33% of the basin area showed an increasing trend in groundwater storage. There were 14 to 33 groundwater drought events in Huaihe basin over the study period, with a gradual increase in drought frequency from north to south across the basin. The duration of groundwater drought showed significant spatial variations with an increasing trend from north to south. Groundwater drought severity exhibited a gradually increasing trend from south to north across the basin, with total severity from 82.77 to 104.40. The variations in groundwater drought indicate the potential risk to environment and water resources management in the Huaihe River Basin.

**Author Contributions:** Conceptualization, B.L. and Z.Z.; methodology, Z.Z. and B.L.; software, Z.Z. and Z.J.; validation, Z.Z., Y.Z. And Z.J.; formal analysis, Z.Z. and B.L.; investigation, Y.Z. and Z.Z.; resources, B.L. and Z.J.; data curation, Z.Z.; writing—original draft preparation, Z.Z.; writing—review and editing, Z.Z. and B.L.; visualization, Z.Z.; supervision, Z.Z.; project administration, B.L. and Y.Z.; funding acquisition, B.L. All authors have read and agreed to the published version of the manuscript.

**Funding:** This research was funded by the National Natural Science Foundation of China (U2240218) and the Science Technology Project of POWERCHINA HUADONG Engineering Corporation Limited (KY2023-HS-02-10).

**Institutional Review Board Statement:** Not applicable.

**Informed Consent Statement:** Not applicable.

**Data Availability Statement:** The raw data supporting the conclusions of this article will be made available by the authors on request.

**Acknowledgments:** Thanks to Teacher Lu Baohong for the guidance of this article, thanks to Jiang Zhengfang and Zhao Yirui for their help in the process of data processing.

**Conflicts of Interest:** The authors declare no conflicts of interest.

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
