# Peer review of "Quantifying Water Storage Changes and Groundwater Drought in the Huaihe River Basin of China Based on GRACE Data"

_sustainability, doi:10.3390/su16198437_

Round 1

Reviewer 1 Report

Comments and Suggestions for Authors

The paper entitled as "Study on Water Storage Changes and Groundwater Drought in the Huaihe River Basin Based on GRACE Data" has a good structure, methodology and data analysis. However, there are some minor comments.

In lines 130 and 131, under section 2.3.1, the following statement is repeated "change; SWESA - change in snow water equivalent; CWSA - change in plant canopy water storage, all in cm; SMSA, SWESA and CWSA are obtained from the GLDAS Noah model." Please, remove it.

In line 147, under section 2.3.3 it is mentioned "Sen et al", reference number 9, although that it is made by only one author. Please, remove et. al.

References from 15 to 27 are not existing in the paper text at all.

Reviewer 2 Report

Comments and Suggestions for Authors Dear Authors,   you have done a good job with the manuscript. However there are many weak points need to be improved before considering the manuscript for publication. I recommend Major revision and here are the comments;   Title: Please add the name of the study area and country in the title and abstract to be clear for the reader  

Abstract:

1. The abstract doesn't show the novelty of the research work and is poorly written.

The author should avoid general sentences and focus only on the current results and finding and add numerical values and show the novelty of the study.   Introduction section: 1. The introduction is too short and it doesn't cover all the topics of the research study. I recommend to increase the literature with comparison with previous work in the study area and global work showing the significance of the methodology of this work. 2. Please add paragraph of the aim of the study  at the end of the introduction 3. I recommend to improve the structure of the introduction section based on the methodology used.   Material and methods section: 1. Study area description (line 68 to 88) Why there are no references? Did the author made all of this work or the information provided from previous work? This point should be clarified. 2. Please add references of previous work in the study area. 3. Figure 1 is not suitable in the current form Please add map of the country  and map of the study area in one figure. The caption is not suitable. I recommend change the caption to be the location map of the study area including (name of river basin).   4. Please add the lat and long on the maps 5. Methods: line 120 to 190 All of the calculation can not be reliable without adding references. I recommend to show the significance of this method through adding or comparison with previous work was done globally or locally using similar method. 6. Figure 2: please rewrite the caption giving more details about the figure.   Results and discussion section: 1.Please compare your results with previous work to strengthen your interpretation. 2. What is the limitations of the research study? 3. Please add paragraph giving recommendations for further research or future work to fill the gap of your research study that couldn't be covered in this study. 4. The discussion section need more improvement.   Conclusion: The author should focus on the current results and avoid generalization of sentences. Please provide quantitative data or numerical values in the conclusions.

References:

The references are not enough and more references to improve the introduction, methodology, results and discussion.

Please correct the manuscript according to my comments! Thank you.

Reviewer 3 Report

Comments and Suggestions for Authors

This paper uses GRACE satellite data and GLDAS-Noah model data to analyze anomalies in terrestrial and surface water storage in the Huaihe River Basin, calculates changes in groundwater storage, and develops a groundwater drought index (GRACE-gdi). The paper is logically clear and applies operational theory to the identification of drought events, using multi-threshold operational analysis for drought recognition. It outlines the trend of changes in terrestrial water storage in the Huaihe River Basin, provides insights into water storage variations, and validates groundwater storage anomalies with measured water level data. This demonstrates GRACE's capability in drought monitoring. However, there are still some errors and suggestions for improvement in the paper:

1.The knowledge gap and a brief overview of the content of this research should be clearly articulated in the introduction section. This will help to set the context and highlight the significance of the study.

2.References should include research conducted by international scholars, particularly those published within the last five years. This will ensure the study is informed by the most current and relevant work in the field.

3.The '3-threshold drought identification method' is mentioned in the research, but as stated in section 2.3.2, this method only identifies drought events without considering their intensity. It is recommended to include additional analysis or discussion on drought intensity to provide a more comprehensive understanding.

4.The calculation of groundwater anomaly storage only presents one method, and the accuracy of the calculation results cannot be verified.

5.The results of the groundwater drought index based on GRACE data are not presented.

6.The term "change rate cm/year" is not precise; it should be expressed as "magnitude of change cm/year."

7.The legend in Figure 7 is not clear enough.

8.The reasons for the correlation between measured groundwater level data and groundwater storage anomalies are not clearly stated.

9.The correlation between groundwater storage changes and measured groundwater levels is not described.

10.The paper mentions that the distributions of drought duration and drought intensity are similar; whether there is a connection between the two is not addressed.

11.The downward trend in both terrestrial water storage and surface water anomalies is noted; whether the reasons for this trend have been considered is unclear.

12.The term "run theory" is miswritten as "tour theory" in the abstract. Please check carefully and correct this mistake.

Comments on the Quality of English Language

The manuscript is generally well-written, but there are instances where the clarity could be improved. The manuscript would benefit from more consistent terminology. I noticed that certain key terms are used interchangeably, which could cause confusion for the reader. Ensuring consistent use of terms throughout the text will enhance the coherence of the manuscript.

Reviewer 4 Report

Comments and Suggestions for Authors

In this study,GRACE satellite data and GLDAS-Noah model data were employed  to analyze terrestrial water and surface water storage anomalies, calculate changes in groundwater storage in the Huaihe River  Basin were calculated, and the groundwater drought index was developed. The results of this study have certain significance for water resources management in river basins, but there are still the following problems that need to be modified, and it is recommended to make  major revision.

1. Please add longitude, latitude, river name, and geographical location in China in Figure 1ï¼›

2. Supplement the specific sources and processing methods of GRACE ,GRACE-FO RL06 Mascon data and GLDAS-Noah data.

3. The data processing of each variable in Formula 1 should be described in detail, How to obtain the change in terrestrial water storage by inversion of GRACE satellite data? How to convert?

4. Table 1 contains less information, so it is recommended to describe it in words.

5. Formula 4 is very simple, but the description is not standard, Qi? N?

6. References to run theory should be supplemented.

7. As can be seen from Figure 3, Figure 4, Figure 5, Figure 6, the accuracy of data is not enough, there is an obvious zigzag graph, and the true result may have been covered up. It is suggested to improve the accuracy of data, re-make the Figures above, and conduct analysis.

8. The paper only analyzes the data in terms of time and space, and lacks discussion links. It is suggested to add discussion content.

9. Water Storage Trends, Spatial Distribution and Rate of Change, and Groundwater Drought Characteristics are summarized, The relationship among them needs to be explained.

Comments on the Quality of English Language

Extensive editing of English language required.

Round 2

Reviewer 2 Report

Comments and Suggestions for Authors

Dear Authors, Thank you for your efforts! I recommend your manuscript in the current form for publication.

Reviewer 4 Report

Comments and Suggestions for Authors

This article has been greatly improved after modification, and it is suggested that it can be published after minor modification.

For example, the river names in Figure 1 should be changed to English.
